# Nonspecific Low Back Pain among Kyokushin Karate Practitioners

**DOI:** 10.3390/medicina57010027

**Published:** 2020-12-30

**Authors:** Wiesław Błach, Bartosz Klimek, Łukasz Rydzik, Pavel Ruzbarsky, Wojciech Czarny, Ireneusz Raś, Tadeusz Ambroży

**Affiliations:** 1Department of Sport, University School of Physical Education, 51-612 Wrocław, Poland; Wieslaw.judo@wp.pl; 2Klimek Fizjoterapia, 31-541 Kraków, Poland; b.klimek599@gmail.com; 3Institute of Sports Sciences, University of Physical Education in Krakow, 31-541 Kraków, Poland; tadek@ambrozy.pl; 4Department of Sports Kinanthropology, Faculty of Sports, Universtiy of Presov, 080-01 Prešov, Slovakia; pavel.ruzbarsky@unipo.sk (P.R.); wojtekczarny@wp.pl (W.C.); 5Doctoral School, University of Physical Education in Kraków, 31-571 Kraków, Poland; ireneuszras@wp.pl

**Keywords:** back pain, Kyokushin karate, training

## Abstract

*Background and objective:* Spinal pain is a common and growing problem, not only in the general population but also among athletes. Lifestyle, occupation, and incorrectly exerted effort have a significant impact on low back pain. To assess the prevalence of low back pain among those practicing Kyokushin karate, we take into account age, body weight, sex, length of karate experience, level of skill, and occupation. *Materials and Methods:* The study involved 100 people practicing Kyokushin karate, aged 18 to 44. A questionnaire developed for this study and the Oswestry Disability Index (ODI) were used. *Results:* The research showed the prevalence of low back pain among karate practitioners (55%), depending on age (R = −0.24; *p* = 0.015), body weight (χ^2^ = 16.7; *p* = 0.002), occupation (χ^2^ = 18.4; *p* = 0.0004), and overall length of karate experience (R = −0.28; *p* = 0.04). A correlation was also found between sex (χ^2^ = 22.3; *p* = 0.001), occupation (χ^2^ = 51; *p* = 0.0000), length of experience (R = −0,28; *p* = 0.04), karate skill level (R = 0.39; *p* = 0.003), and the intensity of pain defined using the Visual Analogue Scale (VAS). Subjects with low back pain showed minimal (71%) and moderate (29%) disability according to the Oswestry index. *Conclusions:* Low back pain is common in karate practitioners and depends on age, weight, occupation, and length of karate experience. The intensity of low back pain is influenced by sex, occupation, overall length of training experience, and one’s level of karate skill. Lumbar spine ailments reduce functionality and quality of life to a small degree. Karate practitioners seldom seek treatment for spinal pains, and only few use physiotherapy and pharmacology.

## 1. Introduction

Spinal pain is now a common and growing problem in modern society. Lumbar and sacral spine pain is experienced by 85% of the adult population at least once in a lifetime, 40% of people experience it once a year, and 15–20% of adults experience it every day. Low back pain syndromes disappear spontaneously within 3 months in 90% of cases; however, they are often recurrent (in 50% of the population) and can develop into a chronic condition [1,2,3]. Scientific studies show that increasing numbers of young people now suffer from low back pain and report that the age at which the condition first appears has fallen, as people aged 20 already suffer from spinal disorders [1,2,3].

Lifestyle, occupation, and incorrectly performed exercise have a significant impact on the increase in spinal pain. Psychosocial factors, such as anxiety, stress, and depression, significantly extend the duration of the ailment and contribute to the transition from an acute to a chronic condition [4,5,6,7].

One’s occupation has a significant impact on the incidence of low back pain. People often spend most of their day at work, and the position they adopt to perform the work, often for extended periods of time, does not have a positive effect on spinal structures. Studies have shown that sitting puts five times more strain on the lumbar spine’s intervertebral structures than standing, and that intense pain often leads to dissatisfaction with one’s job [2,8].

Low back pain is extremely common, not only among the general population but also among athletes [1]. According to Okada et al. [9], 30–85% of all athletes suffer from low back pain. In combat sports, back pain occurs in judokas (according to Okada et al. and Yamaji et al.), Mixed Martial Arts athletes (according to Ridan et al.), and karate practitioners, who are the subject of this study [9,10,11].

Combat sports are characterized by high-intensity and long trainings, and require a high degree of mental concentration, which combined with the frequent shortening of the time needed for tissue regeneration, can place a significant overload on the lumbar spine. The most common causes of lumbar and sacral pain include inadequate biomechanical patterns, muscle imbalance, improper warm-ups, technical errors, and inadequate training, leading to repeated micro-injuries [2,4,10,12,13].

Kyokushin karate is a difficult full-contact sport requiring full engagement, in which trauma usually occurs during intense warm-ups, flexibility exercises with a partner, hitting instruments, and during actual fighting [4]. Studies report that 46% of all karate injuries occur during sparring sessions in training situations, and 13.3% of all karate injuries are back injuries (Ambroży et al.) [12].

The aim of the present study was to assess the incidence of low back pain among those practicing Kyokushin karate in sports clubs in Poland, taking into account age, body weight, sex, overall length of training experience, and karate skill level, as well as occupation.

## 2. Materials and Methods

The study was carried out using a specially designed questionnaire, consisting of two parts—a general and a specific one. The general part included 10 questions—personal questions about age, weight, sex, and field of university study, questions concerning the type of martial arts practiced, and a question qualifying the respondent for the next part of the survey. The specific part consisted of 10 questions concerning the nature of the respondent’s pain, including an assessment of its severity using an analog visual tool, a 10-point Visual Analogue Scale (VAS).

The assessment of life quality in relation to the experienced low back pain was performed using the Oswestry Disability Index [14]. The questionnaire included 10 questions concerning: Pain intensity, personal care, lifting, walking, sitting, standing, sleeping, social life, travelling, and the course of pain symptom changes. Each question had 6 possible answers, for which the respondent received 0 to 5 points. The sum of the points, converted into a percentage, was used to determine the degree of disability: Minimal (0–20%), moderate (21–40%), severe (41–60%), crippled (61–80%), and bed-bound (81–100%).

Grounds for exclusion from the study included: Past injuries and surgeries in the lumbar spine area, traffic accidents, neurological diseases, and lack of desire to take part in the study.

The study was conducted among a group of 100 people training Kyokushin karate in sports clubs in Poland. It involved 26 women and 74 men aged 18 to 32. The body weight of the subjects ranged from 48 to 89 kg for women (average 63.4 kg) and from 60 to 119 kg for men (average 81.1 kg). The subjects were assigned to appropriate weight categories according to the Polish Karate Federation guidelines (Table 1).

The frequency of karate trainings ranged from 1 (21%) to 4–6 trainings per week (9%). Most people trained 2–3 times a week (70%). The duration of the trainings usually ranged between 1 and 1.5 h (57%); 15 people trained from 45 min to 1 h (15%); 13 people for 1 h (13%); 12 people for 1.5 to 2 h (12%); and 3 people over 2 h (3%).

Statistical analysis of the material was done with the use of Statistica 13.1 by StatSoft. Parametric tests were used, which was conditional on meeting basic assumptions of the compatibility of studied distributions to a normal distribution and homogeneity of the variance. The compatibility of the distributions to a normal distribution was assessed with the use of Shapiro–Wilk’s test. The correlation of two variables with a normal distribution was computed using Pearson’s linear correlation coefficient. The statistical significance level was set at *p* < 0.05.

## 3. Results

The study showed that, of the 100 respondents, 55 experienced low back pain, including 42 men and 13 women. In the under-20-years age bracket, 2 people experienced low back pain, in the 20–25 bracket—20 people, in the 26–30 bracket—19 people, and in the over 30 years bracket—14 people. Respondents experiencing low back pain most often were those in the 20–30 age bracket. In the study group, 3 people (5%) in the ≤60 kg weight category for men and ≤50 kg for women complained of low back pain, as did 1 person (2%) in the ≤67 kg weight category for men and ≤55 kg for women; 12 people (22%) in the ≤75 kg category for men and ≤61 kg category for women; 13 people in the ≤84 kg category for men and ≤68 kg category for women; and 26 people (47%) in the above 84 kg category for men and above 68 kg category for women. Noticeably, the highest number of people with low back pain was in the highest weight category. Among the respondents who had trained for less than a year, 2 people (4%) reported low back pain; among those who had trained for 1–2 years—10 people (18%); among those who had trained for 2–5 years—25 people (45%); while 18 people (33%) suffered from pain among those who had trained for over 5 years. Among those at the lowest skill level in karate (8 kyū and below), 14 people (25%) experienced pain, as did 20 people (36%) among those at 7–5 kyū, 18 people (33%) among those at 4–2 kyū, and 3 people (5%) among the most advanced practitioners (Table 2).

Based on the results of the Oswestry Disability Index (ODI), minimal disability was noted in the case of 2 people (5%) in the under-20-years age bracket; among 16 people (41%) in the 20–25 age group; among 12 people (31%) aged 26–30; and among 9 people (23%) over 30. These respondents achieved a score of 0–20% on the Oswestry index. With regard to moderate disability (an ODI score between 21 and 40%), there were 4 people (25%) aged 20–25, 7 people (44%) aged 26–30, and 5 people (31%) aged over 30.

In the group of 37 people (67%) experiencing low back pain, karate training did not affect the occurrence of pain; in 8 people (15%), pain occurred during and after training; in 4 people (7%), pain occurred during training; and in one person (2%) only after training. Finally, 5 people (9%) reported having no more pain after training (Table 3).

Low back pain did not impact the frequency of karate training for 33 people (60%); 16 people (29%) reported some impact on the frequency of their karate training; and pain had a significant impact for 3 people (5%), while 3 others (5%) stopped training karate altogether because of it (Table 4).

Despite experiencing low back pain, most people, that is, 30 people (55%), did not seek treatment; 13 people (24%) used physiotherapy; 5 people (9%) used massages; and 7 people (13%) used pharmacological solutions (Table 5).

A statistically significant positive relationship was found between length of karate training and karate skill level, and pain intensity on the VAS scale. The correlation showed a value of *p* < 0.05. Correlations were weak (R = 0.28 and 0.39 resp). No statistically significant relationship was found between age and weight, and pain intensity. Pain intensity was higher with longer karate training experience and higher karate skill level (Table 6).

## 4. Discussion

Statistical analysis shows that more than half of those training Kyokushin karate in sports clubs in Poland report the occurrence of low back pain. Around 80% of the adult population experiences low back pain at least once in their lifetime. In most cases, low back pain disappears spontaneously; however, it is often recurrent and may develop into a chronic condition [2,3]. In the sports context, these ailments often result from inappropriate biomechanical patterns, inappropriate warm-ups, technical errors, and improper training leading to repeated micro-injuries [10]. Moreover, combat sports and martial arts are characterized by intense training. The length of trainings and the need to retain mental focus, combined with the all-too-frequent shortening of the time needed for tissue regeneration, can place a significant overload on the lumbar spine. Kyokushin karate is a full-contact sport, in which injuries usually occur during overly intense warm-ups, flexibility exercises with a partner, hitting instruments, and during actual fighting [4]. Ambroży et al. [12] have shown that 46% of all karate injuries occur during sparring sessions in training situations, and 13.3% of all karate injuries are back injuries.

The study results show that most of the people experiencing low back pain belonged to the highest weight category. This may be due to body weight, which puts greater strain on the lower part of the back. Studies have shown that 45% of karate practitioners experience quite little pain (3–4 VAS), while 35% report intense pain (7–8 VAS). In a study on the severity of sacral spine pain in students, Stefanowicz et al. [3] showed that the average pain level was 4.1 VAS. Ridan et al. [10] found an average of 3.7 VAS in Mixed Martial Arts (MMA) fighters. This study, examining the intensity of low back pain in karate practitioners, has shown an average pain level of 4.98 VAS, which is higher than in the other cases. A significant relationship has been demonstrated in the statistical analysis between sex and pain intensity. This finding is supported by the research of Sullivan et al. [15], who demonstrated greater resistance and a weaker response to pain among women. This may have to do with the fact that women experience a much higher incidence of pain than men; according to Sundblad et al. [16], more than twice as many people who suffer from pain at least once a week are female. The current study among Kyokushin karate practitioners has shown that practitioners with 2–5 years in training experience represent 45% of all respondents with low back pain. Similarly, Ridan et al. [10] published results according to which Mixed Martial Arts fighters with 1–3 years of experience accounted for 48.4% of all respondents reporting back pain. Thus, it can be concluded that having trained for a relatively short time is a factor that exacerbates injuries because of inappropriate technical habits.

Karate is a full-contact sport requiring full physical and mental attention, in which it is difficult to avoid overload of the spinal structures without knowledge of spinal pain syndrome prevention, anatomy, biomechanics, exercise physiology, and controlled biological renewal.

## 5. Conclusions

Low back pain is common among karate practitioners.Age, weight, and overall length of karate experience all impact the occurrence of low back pain, which is more common in people aged 20–30 with medium-length experience in karate who belong to the highest weight category.Low back pain has little effect on karate practitioners’ functionality and quality of life.Pain intensity assessed using the VAS scale depends on the length of karate training experience, and the practitioner’s karate skill level.

## 6. Applications

Based on the results of the study, we propose introducing McKenzie method’s exercises into the training as a prevention method, allowing the continuation of training despite occurring pain.Studies concerning the impact of these types of exercises on the comfort of the training of karate practitioners are the subject of further analyses; their results will be shown in subsequent papers.

## Figures and Tables

**Table 1 medicina-57-00027-t001:** Number of respondents by weight and sex.

Weight Category	Men	%	Women	%
M ≤ 60 kg; W ≤ 50 kg	1	1%	3	12%
M ≤ 67 kg; W ≤ 55 kg	8	11%	1	4%
M ≤ 75 kg; W ≤ 61 kg	21	28%	9	35%
M ≤ 84 kg; W ≤ 68 kg	16	22%	7	27%
M + 84 kg; W + 68 kg	28	38%	6	23%
Total	74	100%	26	100%

M—mean, W—woman.

**Table 2 medicina-57-00027-t002:** Occurrence of low back pain according to selected features.

Features		People Experiencing Pain	People without Pain	*p* χ^2^ Pearson
Sex	Men	42	32	0.551
	Women	13	13	
Age Bracket	<20 years old	2	6	0.015
	20–25	20	23	
	26–30	19	9	
	>30	14	7	
Weight Category	M ≤ 60 kg; W ≤ 50 kg	3	1	0.002
	M ≤ 67 kg; W ≤ 55 kg	1	8	
	M ≤ 75 kg; W ≤ 61 kg	12	18	
	M ≤ 84 kg; W ≤ 68 kg	13	10	
	M + 84 kg; W + 68 kg	26	8	
Overall Length of Training Experience	<1 year	2	8	0.016
	1–2 years	10	10	
	2–5 years	25	19	
	5 years or more	18	8	
Karate Skill Level	8 kyū and below	14	16	0.223
	7–5 kyū	20	18	
	4–2 kyū	18	6	
	1 kyū and above	3	5	

*p*—probability value, Source: Own study. Bold indicates that the result is statistically significant.

**Table 3 medicina-57-00027-t003:** Impact of karate training on low back pain.

Training and Variability of Pain	People Experiencing Pain
Experiencing pain during training	4
Experiencing pain after training	1
Pain stops after training	5
Pain occurs during and after training	8
No impact	37
Total	55

**Table 4 medicina-57-00027-t004:** Impact of pain on training frequency.

Impact of Pain on Training Frequency	People Experiencing Pain
No impact	33
Some impact	16
Considerable impact	3
Stopped training	3
Total	55

**Table 5 medicina-57-00027-t005:** Treatment sought by respondents.

Type of Treatment	People Experiencing Pain
No treatment	30
Physiotherapy	13
Massage	5
Analgesic and anti-inflammatory medicine	7
Total	55

**Table 6 medicina-57-00027-t006:** Assessment of correlation with pain intensity.

Variable	R	*p*
Age and pain intensity	0.03	0.776
Weight and pain intensity	0.25	0.069
Length of karate training experience and pain intensity	0.28	0.041
Karate skill level and pain intensity	0.39	0.003

R—Spearman’s rank correlation coefficient; *p*—probability value, Source: Own study. Bold indicates that the result is statistically significant.

## Data Availability

The data presented in this study are available on request from the corresponding author.

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
