# Peer review of "Nonspecific Low Back Pain among Kyokushin Karate Practitioners"

_medicina, 2020, doi:10.3390/medicina57010027_

Round 1
Reviewer 1 Report
This study was a cross sectional study of the low back pain (LBP) of Kyokushin karate practitioners and analyzed the correlation between various variables including age, weight, occupation, and length of karate experience and LBP.
In this study, various outcome measures for evaluating pain such as the presence or absence of pain, VAS, and Low Back Pain Oswestry questionnaire (maybe, ODI), were used, but it is difficult to conclude that the correlation between each variable and each pain indicator is consistent.
Especially, the result that the intensity of pain has a high correlation according to age, weight, occupation, and length of karate experience was not sufficiently explained in the pain intensity measured by VAS.
In addition to the p-value, careful interpretation of the correlation coefficient was required.
It is questionable whether the sample size of 100 people was sufficient for analysis.
Detailed information on the statistical analysis method was omitted in the method.
Minor comments:
Correct use of abbreviations is required. 'LBP' is often used as an abbreviation for low back pain. The full names of 'MMA' and 'VAS' did not appear throughout the manuscript.
A reference to the 'Oswestry Low Back Pain Questionnaire' should be added. I think this questionnaire refers to the Oswestry Disability Index (ODI). It must be clearly confirmed.
Author Response
Dear Reviewer,
Thank you very much for your time and valuable comments, which all have been considered and incorporated. The detailed list of responses is given below. We hope that the modifications and explanation will be acceptable for you.
Yours sincerely,
Rydzik, corresponding author
This study was a cross sectional study of the low back pain (LBP) of Kyokushin karate practitioners and analyzed the correlation between various variables including age, weight, occupation, and length of karate experience and LBP.
In this study, various outcome measures for evaluating pain such as the presence or absence of pain, VAS, and Low Back Pain Oswestry questionnaire (maybe, ODI), were used, but it is difficult to conclude that the correlation between each variable and each pain indicator is consistent.
A: The occurrence of pain correlated with sometimes different parameters (age, category etc.) than the intensity of pain. The results could be more consistent with larger number of participants.
Especially, the result that the intensity of pain has a high correlation according to age, weight, occupation, and length of karate experience was not sufficiently explained in the pain intensity measured by VAS.
A: corrected
In addition to the p-value, careful interpretation of the correlation coefficient was required.
A: corrected
Detailed information on the statistical analysis method was omitted in the method.
A : corrected
Correct use of abbreviations is required. 'LBP' is often used as an abbreviation for low back pain. The full names of 'MMA' and 'VAS' did not appear throughout the manuscript.
A: corrected
A reference to the 'Oswestry Low Back Pain Questionnaire' should be added. I think this questionnaire refers to the Oswestry Disability Index (ODI). It must be clearly confirmed.
A : Changed to ODI, added item 13 on the reference list
Reviewer 2 Report
Although this research could be interesting and it is a well written and designed work, I do not think that this paper might strongly contribute to increase the knowledge about the topic. In other words, we know from the beginning that this group of population is at risk for low back pain. I think that could be interesting to focus the attention of the readers about the ability of the karate practioners to continue their activities, despite the low back pain. So my suggestion to the authors, is to highlight this aspects.
Author Response
Dear Reviewer,
Thank you very much for your time and valuable comments, which all have been considered and incorporated. The detailed list of responses is given below. We hope that the modifications and explanation will be acceptable for you.
Yours sincerely,
Rydzik, corresponding author
Although this research could be interesting and it is a well written and designed work, I do not think that this paper might strongly contribute to increase the knowledge about the topic. In other words, we know from the beginning that this group of population is at risk for low back pain. I think that could be interesting to focus the attention of the readers about the ability of the karate practitioners to continue their activities, despite the low back pain. So my suggestion to the authors, is to highlight this aspects.
A: Additions in applications were made that highlight the aspects.